Flower, fruit phenology and flower traits in Cordia boissieri (Boraginaceae) from northeastern Mexico

Martínez-Adriano Cristian Adrian 1
Jurado Enrique 1 enrique_jurado@hotmail.com
Flores Joel 2
González-Rodríguez Humberto 1
Cuéllar-Rodríguez Gerardo 1
1 Facultad de Ciencias Forestales, Universidad Autonoma de Nuevo Leon , Linares Nuevo Leon , Mexico
2 Ciencias Ambientales, Instituto Potosino de Investigación Científica y Tecnologica , San Luis Potosi San Luis Potosi , Mexico
Ross-Ibarra Jeffrey
Electronic publication date: 2016 May 17
Publication date: 2016
Volume: 4
Electronic Location ID: e2033
Received 2015 Nov 19; Accepted 2016 Apr 21
Copyright: ©2016 Martínez-Adriano et al.
Copyright year: 2016
Copyright holder: Martínez-Adriano et al.
License: This is an open access article distributed under the terms of the Creative Commons Attribution License, which permits unrestricted use, distribution, reproduction and adaptation in any medium and for any purpose provided that it is properly attributed. For attribution, the original author(s), title, publication source (PeerJ) and either DOI or URL of the article must be cited.
License URL: https://creativecommons.org/licenses/by/4.0/

Keywords: Anacahuita, Floral dimorphism, Tamaulipan thornscrub, Heterostyly, Rainfall

Funding: CONACYT 230073 PAICYT UANL This research was partially supported by a grant from CONACYT 230073 and PAICYT UANL. The funders had no role in study design, data collection and analysis, decision to publish, or preparation of the manuscript.

==============================
We characterized variations in Cordia boissieri flowers and established if these variations occur between plants or between flowering events. Flowering and fruiting was measured for 256 plants. A GLM test was used to determine the relationship between flowering and fruit set processes and rainfall. We performed measurements of floral traits to detect variations within the population and between flowering events. The position of the anthers with respect to the ovary was determined in 1,500 flowers. Three out of four flowering events of >80% C. boissieri plants occurred after rainfall events. Only one flowering event occurred in a drought. Most plants flowered at least twice a year. The overlapping of flowering and fruiting only occurred after rainfall. Anthesis lasted three-to-five days, and there were two flower morphs. Half of the plants had longistylus and half had brevistylus flowers. Anacahuita flower in our study had 1–4 styles; 2–9 stamens; 6.5–41.5 mm long corolla; sepals from 4.5–29.5 mm in length; a total length from 15.5–59 mm; a corolla diameter from 10.5–77 mm. The nectar guide had a diameter from 5–30.5 mm; 4–9 lobes; and 5 distinguishable nectar guide colors. The highest variation of phenotypic expression was observed between plants.

Introduction

Phenology is defined as the study of the seasonal and recurrent timing of life cycle events (Rathcke & Lacey, 1985; Williams-Linera, 2003) influenced by climate and weather (Dubé, Perry & Vittum, 1984; Newstrom, Frankie & Baker, 1994; Schwartz, 2003; Richardson et al., 2013). Examples of this common phenomenon are migration and hibernation in animals, or sprouting, leaf expansion, abscission, flowering, fertilization, seed set, fruiting, seed dispersal and germination in plants (Dubé, Perry & Vittum, 1984; Wright & Cornejo, 1990; Fenner, 1998; Elzinga et al., 2007; Lediuk et al., 2014).

For plant phenology, changes in factors like humidity (Reich & Borchert, 1984; Barone, 1998; Hódar, Obeso & Zamora, 2009; Silveira, Martins & Araújo, 2013), temperature (Reich & Borchert, 1984; Justiniano & Fredericksen, 2000; Yadav & Yadav, 2008), photoperiod (Reich & Borchert, 1984; Barone, 1998), edaphic associations and topography (Sánchez-Azofeifa et al., 2003) strongly influence reproduction and population survival (Rathcke & Lacey, 1985; Elzinga et al., 2007). Biotic factors, such as competition for pollinators (Janzen, 1967; Gentry, 1974; Murray et al., 1987; Sánchez-Azofeifa et al., 2003; Trapnell & Hamrick, 2006), frugivory, seed dispersal (Rathcke & Lacey, 1985; Chapman et al., 2005) and herbivory (Williams-Linera, 1997) may shape plant phenology. Thus, phenology patterns displayed by plants are adaptations to the surrounding abiotic and biotic environments (Rathcke & Lacey, 1985; Van Schaik, Terborgh & Wright, 1993; Williams-Linera, 2003; Elzinga et al., 2007).

Floral traits are selected to ensure sexual reproduction, because cross-fertilization increases genetic variability which is advantageous to offspring (Charlesworth & Charlesworth, 1987). Many traits are involved in attracting pollen vectors, such as flower size, shape, color and scent (Hargreaves, Harder & Johnson, 2009; Rosas-Guerrero et al., 2011). Plant species commonly show plastic traits, because the pollinator composition and abundance vary within and between reproductive seasons (Harder & Johnson, 2005). If floral visitors favor a floral trait within a population, that could lead to a divergence in floral phenotype (Sánchez-Lafuente, 2002; Huang & Fenster, 2007; Brothers & Atwell, 2014). The timing of plant reproductive cycles affects not only plants but also animals that depend on plant resources (Newstrom, Frankie & Baker, 1994).

Species with floral morphology associated with specialized pollination systems have less variation within populations than those species with attributes associated with generalized pollination systems (Stebbins, 1970; Ushimaru, Watanabe & Nakata, 2007; Van Kleunen et al., 2008; Rosas-Guerrero et al., 2011). Studies in phenotype selection have been mainly focused on evaluating the effect of floral attributes on plant fitness; each flower is a complex unit with a configuration for an appropriate function (Ushimaru, Watanabe & Nakata, 2007; Armbruster & Muchhala, 2009; Rosas-Guerrero et al., 2011). This configuration can produce adaptations of flower traits with pollinators as often underestimated selective agents (Dilcher, 2000; Ushimaru, Watanabe & Nakata, 2007; Rosas-Guerrero et al., 2011). One of these adaptations is heterostyly, a floral polymorphism in style and stamen length (Gasparino & Barros, 2009). The phenomenon occurs in the same species, in two (distyly) or three forms (tristyly) (Darwin, 1877; Faivre & Mcdade, 2001; Gasparino & Barros, 2009). Heterostyly occurs in many plant families, a dimorphism in style length (Opler, Baker & Frankie, 1975) was documented in Boraginaceae (i.e., Cordia) by Fritz Muller, when he sent samples from Brazil to Charles Darwin (Darwin, 1877). Heterostyly has evolved at least 12 times in Boraginaceae (the largest number of origins in any family) (Cohen, 2014) and is present in at least nine genera (Naiki, 2012). The descriptions of flower morphs found for Cordia describe heterostyly for at least 9 of the 250 or more species (Opler, Baker & Frankie, 1975; McMullen, 2012; Naiki, 2012; Canché-Collí & Canto, 2014).

In Cordia, flowering time tends to vary in relation to moisture availability: northern Cordia species begin flowering in regions with severe drought, earlier than those from areas with less water stress (Borchert, 1996). Anacahuita (Cordia boissieri A. DC.) occurs in Nuevo León, Tamaulipas, Veracruz, Hidalgo and San Luis Potosí Mexico and South Texas in the US (Gilman & Watson, 1993; Alvarado et al., 2004), it is the state flower of the Mexican state of Nuevo León, and its flowering has not been studied. Research on other Cordia species (Opler, Baker & Frankie, 1975; De Stapf & dos Santos Silva, 2013) found a wide range of reproductive systems, ranging from the homostylous to heterostylous and dioecious, including those adapted for pollination assemblages, with both wind and animals as pollen vectors. Cordia alliodora and C. elaeagnoides, for instance, bloom at the end of the rainy season; however C. alliodora blooms were found to occur later and to last longer (Bullock & Solis-Magallanes, 1990). The blooms of C. glabra occur from August to September in a monoecious reproductive system (Justiniano & Fredericksen, 2000). In contrast C. multispicata produces flowers and fruits during most of the year, with peak flowering between the end of the dry season and half way into the wet season (Vieira & Silva, 1997). The anthesis of this species lasts up to six days.

We characterized the phenological variation in Anacahuita flowers and established if this variation occurs in the same flowering event or between different events, considering the following questions: is there a relationship between flowering and fruiting with rainfall events? Is there a variation of Anacahuita floral traits between plants or between flowering events? Plant species that flower more than once a year could present differences in flowers if they target different species of pollinators (Bawa et al., 1989). The most common families found as potential pollinators of Cordia boissieri in our study site were Coleoptera: Scarabidae, Hymenoptera: Apidae, Formicidae, Lepidoptera: Hesperiidae, Nymphalydae, Papilionidae, Pieridae and Sphingidae during the day and Hymenoptera: Formicidae and Lepidoptera: Sphingidae during the night (Martínez-Adriano, 2011). Two families were active most of the year (Hymenoptera: Apidae and Formicidae) while others were seasonal (Martínez-Adriano, 2011), as found in other studies of pollinators (Martín-González et al., 2009). Another hypothesis is that pollination, seed dispersal, or seed germination may increase with several flowering events rather than one extended event (Bullock, Beach & Bawa, 1983), because the opportunities for seedling establishment in the study area can occur in more than one season perhaps with similar chances of success (Jurado et al., 2006).

Materials and Methods

Study area

The study was performed from October 2009 to September 2011. We worked with a population of 256 individuals of Anacahuita in a fragment of Tamaulipan thornscrub in Northeast Mexico, inside the experimental area of Facultad de Ciencias Forestales (Universidad Autónoma de Nuevo León, 24°46′43″N99°31′39″W) at an elevation of 370 m above sea level. The area has an average temperature of 21 °C, with a maximum extreme temperature in summer >40 °C and <0 °C in winter. The annual rainfall average is 805 mm, and dominant soils in the area are vertisols of alluvial-colluvial origin (SPP–INEGI, 1986; Alanís-Rodríguez et al., 2008).

Species description

Cordia boissieri is a native North American shrub or small tree, 5 to 8 m tall. It has simple, alternate and ovate leaves from 15 to 20 cm in length, with a pilose-velvety surface (Vines, 1986; Gilman & Watson, 1993). The flowers are trumpet shaped, in groups from five to eight, white with a yellow nectar guide, up to 45 mm in length, with five rounded lobes and five stamens joined at the base within the corolla tube (Vines, 1986; Gilman & Watson, 1993; Alvarado et al., 2004). In addition, the anthers are oblong, filiform, glabrous, and yellow–greenish; the pistil usually varies in length and narrows towards the apex, ending with two stigmas. Flowering occurs throughout the year, with peaks in the rainy season from late spring to early summer (Vines, 1986; Gilman & Watson, 1993; Alvarado et al., 2004).

Prospective visits were made in order to identify and mark all individuals >1.5 m in height. Each individual was marked with a metallic label and flagging tape with progressive numbers. Five plants were randomly selected to follow the life of 10 flowers in each one, from bud opening to flower senescence, in order to determine flower life span and avoid duplicating data during subsequent samplings.

Flowering and rainfall

In order to determine variations on flower and fruit production of Anacahuita through time and how it relates to rainfall, we recorded flowering of individuals within the population and the monthly amount of rainfall (mm) in the study area. During the study we quantified monthly: highly flowering plants (as seen below), plants with fruits, and plants without flowers and fruits. Flowering in Anacahuita showed two very distinct patterns, one was a very dense flower production that covers the entire canopy and the other is the production of a few flowers in one or two branches (Fig. 1). Only plants with dense flower cover were considered as highly flowering plants. The same criterion was used for fruit production. To determine the relationship between highly flowering plants and fruiting with rainfall events, we performed a GLM analysis using STATS package for R software (R Core Team, 2014), where the dependent variables were plants with flowers and plants with fruits and the independent variable was rainfall. We considered rainfall of one month prior to flowering and two months prior to fruit set following our observations that flowering and fruiting roughly occur after this time.

Figure 1 Examples of Cordia boissieri plants with distinct flower cover.

Plant without or with very few flowers (A) and plant with high numbers of flowers (B). Only those with high dense flower cover were considered as flowering.

Flower measurements

To determine phenotypic variation of flowers within plants and between plants and in different flowering events, we selected five flowers at random in each cardinal point, obtaining data from 20 flowers for each one of five plants during each flowering event. The floral attributes evaluated were as follows: Whether flowers were longistylus or brevistylus (style length) (ST); style number (SN); stamens number (SAN); total flower length (TL; mm with a 0.05 mm accuracy); corolla length (CL; mm); corolla diameter (CD; mm); nectar guide diameter (NGD; mm); number of lobes (NL); nectar guide color (NGC; by direct observation of changes in patterns and color tones in the flower tube) and sepal length (SL; mm), obtained through the difference of the total and corolla lengths. Style length was determined in 1,500 flowers from 75 plants across flowering events (>5% of high flowering plants). To reduce color bias due to perception, this trait was assessed by only one person in full day light against a white sheet of paper.

Recorded data were tested for normal distribution using Kolmogorov–Smirnov test. We performed a nested ANOVAs to test for differences in floral traits among plants and flowering events (P = 0.05) for quantitative variables. Kruskal-Wallis tests were performed for categorical and variables not normally distributed (Zar, 2010). We used a binomial test for style length and goodness of fit using χ2 tests (Ríus-Díaz et al., 1998) for nectar guide colors to determine if there were variations in the phenotypic expressions of the studied plants.

Figure 2 Fluctuation of flowering and fruiting of Cordia boissieri plants and rainfall.

Flowering generally resulted after one month of rainfall and fruiting after two months. Four flowering events included many plants (>80% ). These had a longer duration, we observed an overlapping of flowering and fruit production in three of the flowering events, flower and fruit set were observed as separate processes in the other flowering event with many plants (April 2011).

Results

Flowering and rainfall events

C. boissieri flowering generally resulted after one month of rainfall and fruiting after two months (Fig. 2); the GLM analyses showed a significant relationship between rainfall events and the number of plants with flowers (F1,22 = 817.33; P = 0.003) and fruits (F1,22 = 143.13; P < 0.001). We recorded nine flowering events, one in 2009 (October), five in 2010 (April, June, July, August, September and October) and three in 2011 (April and July), we also observed isolated flowering plants in November 2009 (three plants), March 2010 (six plants), and March 2011 (five plants). Four flowering events included over 80% of plants (215). These had a longer duration (Fig. 2).

We observed an overlapping of flowering and fruit production in three of the four flowering events with >80% of the plants, flower and fruit set were observed as separate processes in the other flowering event with >80 of the plants with many flowers (April 2011) (Fig. 2). The latter was the only event that occurred after no rainfall.

Flower trait variation

We determined that anthesis (opening to senescence of flowers) of C. boissieri was from three to five days. We observed heterostyly on C. boissieri flowers (Supplemental Information 1): there were 760 brevistylus, flowers (from 38 plants) and 740 longistylus flowers (from 37 plants). The style length did not vary within individuals; however, the Kruskal-Wallis test showed that there was a significant variation between plants (H1,74 = 1, 499; P < 0.001). In addition, the binomial test for style length showed similar numbers longistylus (51%) and brevistylus flowers (49%; P = 0.624). Style length and nectar guide color (NGC) did not vary between flowering events.

The phenotypic expression most commonly observed was two styles, five stamens and five lobes. However, in 2,267 observed C. boissieri flowers we found flowers from one to four styles (x ¯=2.02±0.135 SD), with two to nine stamens (x ¯=5±0.381 SD) and two to nine lobes (x ¯=5.01±0.381 SD, as seen in Supplemental Information 2). We found five nectar guide colors (Supplemental Information 3), in the 1,500 flowers from 75 plants. There were 760 flowers with yellow, 380 with yellow-orange, 260 with yellow-white, 60 with yellow-white-orange and 40 with an orange-yellow nectar guide. These numbers were significantly different to chance variations (χ2 = 1149.333; df = 4; P < 0.001), and varied between plants (H1,74 = 1, 499; P < 0.001).

Cordia boissieri flowers ranged in total length from 15.45 mm to 58.9 mm (x ¯=35.49±5.78 SD), corolla varied from 6.45 mm to 41.1 mm (x ¯=24.08±5.26 SD), and sepal from 4.45 mm to 29.5 mm (x ¯=11.4±2.57 SD). Corolla diameter varied from 16.3 mm to 77.0 mm (x ¯=47.98±9.33 SD), whereas nectar guide diameter ranged from 4.9 to 30.6 mm (x ¯=11.64±2.22 SD). Nested ANOVAs for each floral attribute showed that most variations occurred between plants than between flowering events. Total flower length (F2,81 = 18.217, P < 0.001), corolla length (F2,81 = 14.532, P < 0.001), corolla diameter (F2,81 = 47.316, P < 0.001) and nectar guide diameter (F2,81 = 42.335, P < 0.001) differed between plants but not between flowering events. Sepal length varied both between plants (F2,81 = 5.697, P < 0.001) and between flowering events (F81,1596 = 3.156, P = 0.048).

There were significant variations between plants for style length (H2,27 = 1679.000, P < 0.001), style number (H2,27 = 112.868, P < 0.001), number of anthers (H2,27 = 126.521, P < 0.001), number of lobes (H2,27 = 143.071, P < 0.001), and color of nectar guide (H2,27 = 1679.000, P < 0.001). Number of lobes also differed between flowering events (H2,27 = 15.618, P < 0.001). No intra plant variations in floral traits were detected comparing the four sides (cardinal points) of the canopies (P > 0.05).

Discussion

Flowering and rainfall events

The phenological characteristics of flowering (intensity, duration and overlap) are important aspects of the reproductive effort of plant (Richards, 1986; Guitián & Sánchez, 1992), and their variation (including opening time and number of flowers) commonly correlates with temperature, moisture, and day length (Justiniano & Fredericksen, 2000; Elzinga et al., 2007; Yadav & Yadav, 2008; Hódar, Obeso & Zamora, 2009); edaphic and biotic factors (Borchert, 1983) like pollinators (Trapnell & Hamrick, 2006; Medel & Nattero, 2009) seed dispersers and herbivores (Mahoro, 2002; Lacey et al., 2003). C. boissieri is described in the literature as producing flowers and fruits all year, with two peaks of blooms in late spring and early summer (Vines, 1986; Gilman & Watson, 1993; Alvarado et al., 2004); this may be a result of a bimodal rainfall pattern (Chapman et al., 1999). Our results agree with the literature, because the plants of Cordia boissieri showed four flowering peaks during the two years of observation and were approximately in late spring and early summer. The general coincidences of flower traits across time suggest that the species might be pollinated by the same agents.

We observed that flowering was more abundant one month after heavy rainfall events, but also one abundant flowering event occurred after a dry season. This flowering coincides with reports form the literature of highest blooming events in plants of an arid tropical community during the dry and warm season (León De La Luz, Coria–Benet & Cruz–Estrada, 1996). The common pattern for angiosperms is one annual flower production and only for some plants, flowering may occur more than once a year to the point that some species produce flowers and fruits all year (Wolfe & Burns, 2001). The variation in time of year for flowering indicates that unlike for other studies (i.e., Borchert et al., 2005) the variation of moisture is very important in triggering flowering in this C. boissieri population. The observed flowering events of C. boissieri could be induced by previous rainfall events that may ensure seed production (Elzinga et al., 2007). Our results for Cordia boissieri agree with those observed in Cordia alliodora and C. elaeagnoides that bloom at the end of the rainy season (Bullock & Solis-Magallanes, 1990) and were similar to C. multispicata that is described as producing flowers and fruits during most of the year (Vieira & Silva, 1997). Further long-term studies might consider temperature and rainfall or perhaps evapotranspiration or other water availability indicator as a predictor of flower and fruit production.

Flower trait variation

We observed that Anacahuita flowers had a lifespan of five days, this may be due to the receptivity of flowers, because the time to anthesis of each species varies with pollination systems; for example there are plants that have flowers that are receptive from only a few hours (Armbruster & Muchhala, 2009) to more than two weeks (Steinacher & Wagner, 2010). C. boissieri showed a diurnal and nocturnal display, suggesting diurnal and nocturnal pollinators. C. boissieri flowers showed a heterostyly pattern, belonging to the group of distyly flowers, coinciding with previous findings for Cordia to avoid self-pollinating (Gibbs & Taroda, 1983; McMullen, 2012). Boraginaceae is one of the 24 families that exhibit heterostyly, with the Cordia species displaying distyly and tristyly (Ganders, 1979).

Previous studies indicate that it could be misleading to characterize the phenotype of a single flower, because there could be an intra-plant floral variation, which may even be greater than between plants within populations (Medel & Nattero, 2009). In contrast, floral traits measured in our study showed more variation between plants than within the same plant or between flowering events.

The observed flowers in this study allowed us to find some differences from the flower description for the species. For example, in terms of styles, the general morphotype or most dominant is of two styles per flower (Vines, 1986; Gilman & Watson, 1993; Alvarado et al., 2004); however; we found flowers with one to four styles. Anacahuita flowers are described as having five stamens and five lobes with a length of 45 mm, with flowers with a yellow nectar guide (Vines, 1986; Gilman & Watson, 1993; Alvarado et al., 2004); in contrast, we found a flower length from 15.45 mm to 58.9 mm and five phenotypic expressions of nectar guide color, including the yellow variation cited in the literature. This could be considered for a new description of C. boissieri flowers.

In conclusion, three of the four flowering events with >80% of C. boissieri plants occurred one month after heavy rainfall events. Fruit set occurred after two months of rainfall. The species showed overlapping of flowering and fruiting only after heavy rainfall events but not during the dry season. The highest variation of phenotypic expression of C. boissieri flowers occurred between plants and not between flowering events. We suggest considering new flower phenotypes described here for future flower descriptions of C. boissieri. Further longer term studies may help understand if variations in flower traits are the result of similar insects pollinating Anacahuita flowers throughout the year.

Supplemental Information

Supplemental Information 1 Raw data used for analyses from Cordia fruits and flowers in NE Mexico

pdf file with raw data with explicit headings

Click here for additional data file.

Supplemental Information 2 Morphs of Cordia boissieri flowers

Arrows are parallel to the ovaries. S = superior style, I = inferior style. The stamens are highlighted inside dotted red lines. From 15,000 flowers (from 75 plants), we found 760 to have an inferior ovary (from 38 plants) and 740 to have a superior ovary (from 37 plants).

Click here for additional data file.

Supplemental Information 3 Example of variation on number of lobes of Cordia boissieri flowers

Click here for additional data file.

Supplemental Information 4 Nectar guide colors of Cordia boissieri flowers

Click here for additional data file.

The authors thank JM López, E Fernández, E Zaragoza, B Soto, D Peñaflor, P Hinojosa, M Garza, D Salas, and JA López for fieldwork assistance. M Wilson edited English grammar and style. J Ross-Ibarra and an anonymous reviewer greatly improved the clarity of our manuscript.

Additional Information and Declarations

Competing Interests

Author Contributions

Data Availability

The authors declare there are no competing interests.

Cristian Adrian Martínez-Adriano and Enrique Jurado conceived and designed the experiments, performed the experiments, analyzed the data, contributed reagents/materials/analysis tools, wrote the paper, prepared figures and/or tables, reviewed drafts of the paper.

Joel Flores conceived and designed the experiments, analyzed the data, wrote the paper, prepared figures and/or tables, reviewed drafts of the paper.

Humberto González-Rodríguez conceived and designed the experiments, contributed reagents/materials/analysis tools, wrote the paper, reviewed drafts of the paper.

Gerardo Cuéllar-Rodríguez conceived and designed the experiments, performed the experiments, analyzed the data, contributed reagents/materials/analysis tools, wrote the paper, reviewed drafts of the paper.

The following information was supplied regarding data availability:

The raw data has been supplied as Supplemental Information 1.

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
