# Peer review of "Flower, fruit phenology and flower traits in Cordia boissieri (Boraginaceae) from northeastern Mexico"

_PeerJ, doi:10.7717/peerj.2033_

## Round 0.1 · original submission · Major Revisions

· Academic Editor

Major Revisions

Your manuscript has been reviewed by myself and one expert reviewer. While the reviewer is very critical of the manuscript, I suspect that sufficient revisions could address many of these concerns. I share a number of these, including:

It is not clear to me what "Only 127 plants with abundant (i.e >50%) flower or fruit cover were considered." means. Only those plants were ever considered? Things with <50% flower cover were not counted as flowering? How is percentage flower cover calculated?

"We performed nested ANOVAs to test for differences in floral traits among the population"... perhaps this is a language issue, but as far as I can tell there is only one population? How was the nesting done?

I also agree with the reviewer that some additional analyses would help. It appears the regression analyses were done only on precipitation from the previous month. Do we have good reason to expect that is the most meaningful? Why not 2 months ago, or 6 months? Further justification or exploration of the data is warranted I think.

I also agree with the reviewer further explanation of methods is needed, e.g. what indeed is the integration index?

The y-axis on figure 1 is confusing: "flower-fruit" should say "No flowers or fruit".

There are a number of places where the writing is confusing. For example: "In our study we found that most of the variation of floral attributes measured was between plants and not between blooms or within a plant, except for color (C) and style type (TE) which did not vary between blooms or within a plant." If I'm reading this right, it sounds like the variation in C and TE must be among plants since it is not between blooms or within a plant -- so how does that make them different from other traits? Careful reading of the English and ensuring that a reader not familiar with the subject can follow the logic is suggested.

Reviewer 1 ·

Basic reporting

The Introduction is unclearly written and fails to convey the hypotheses to be tested. The introduction oversimplifies the proximate factors that shape flower morphology and plasticity. For example the authors claim that animal pollinated flowers are more likely to show plastic variation due to changes in pollinator abundance, but fail to present examples that support their claim. The authors seem to equate phenotypic variation with plasticity, which is incorrect. The main objective of the manuscript is to characterize phenological variation of this species within or between “blooming processes”, however it is unclear what the authors mean by “blooming process” or why there should be variation between them. The authors also propose to test differences in floral traits between flowering events, however the introduction never explains why would one expect such differences. Therefore, the introduction fails to justify the study.

This manuscript needs to be revised thoroughly and analysis need to be expanded and redone. The language needs to be tightened significantly, several sentences are unclear and there are multiple terms that are mentioned but never explained.
Figures 2-4 should be supplemental.

Experimental design

There are also several problems with the methods section. The study is based on a single population, therefore generalizations based on flower morphology and intra-seasonal variation in flowering phenology may be biased. Phenology tracking is not explained. Information on the timing and periodicity of phenology data recordings is missing. The authors also fail to explain how is flowering intensity recorded or what criteria are used to define a blooming event. It appears that plants are considered to be flowering or not, however flowers may last up to three days. Flowering intensity and duration should be taken into account to determine if rainfall is a proximate factor causing flowering. The statistical analysis should be clarified further and revised. After reading the manuscript I still cannot understand if rainfall causes immediate flowering or flowering occurs after a certain amount of time after rainfall (which is usually the case, e.g. Erythroxylum). Generalized linear models (or GLMM) are probably more appropriate for their data.

FLower measurements were compared among populations, but the study was conducted on a single site, therefore it is unclear what the authors are testing. Random variation within and between individuals is not considered in their analysis, therefore the study suffers from pseudoreplication. In the results section they later mention that they used a nested anova, which was not explained in the methods section. “Integration of flower traits” is never detailed. It is unclear what this is and why should it be calculated? Based on the results section it appears that flower trait integration is basically correlation analysis between variables. Why not simply call them that?

Validity of the findings

The discussion reaches several conclusions that are not fully supported by the data analysis. The relationship between flowering and rainfall may depend on other factors (as mentioned in the discussion) which were not tested or analyzed. Variation in flower morphology are not discussed at all. Several claims, for examples that angiosperms (all of them) are more likely to be annuals or that pollinators do not “shape” flower morphology in generalist, need to be revised against current literature. Conclusions summarise results and did not present any interpretation of the hypotheses.

---

## Round 0.2 · Minor Revisions

· Academic Editor

Minor Revisions

Thank your for your attention to reviewers' concerns. Reviewer #1 still believes some additional changes are warranted, however. If you feel you can address the reviewer's concerns, especially regarding statistical reporting and additional information required (e.g. pollinators), I believe the paper could be acceptable for publication and will not need any additional review.

Reviewer 1 ·

Basic reporting

The manuscript has certainly improved since the last version, however language is still ambiguous in some areas (detailed in general comments) particularly in the methods section. The introduction fails to support the hypotheses to be tested. The link between morphology and phenology could be clarified by providing information on pollinators, however I strongly suggest the entire focus of the paper should be shifted towards a descriptive analysis.

This study describes the reproductive phenology and flower morphology of one population of Cordia boissieri in Mexico. In the introduction the authors propose to test how changes in phenology may reflect changes in flower morphology. Changes in pollinator assemblages are proposed as the proximate cause affecting flower shape for individuals with different flowering schedules. However a major problem with this study is the complete lack of information regarding pollinators. The authors do not inform us about the possible suite of pollinators or visitors, their relative abundance and how it changes through time. Therefore it is impossible to determine if the proposed hypothesis is valid or just a straw man. I propose that the introduction may be reframed as presenting basic descriptive information on a rarely studied species which may prove useful for future studies. In its present form, the introduction does not provide sufficient context for the study nor justifies the research questions.

Figures are relevant however Figure 2 could be improved. The y-axis may be changed to display the relative frequency (percentage) of the study population on each phenophase, thus making the black line unnecessary improving the clarity of the graph.

Experimental design

The authors do not communicate how this research may fill an identifiable knowledge gap. As stated previously, if the manuscript is reframed as a descriptive study of a poorly studied species, that should place the research into context.

I still need to see some more detail on statistical analyses. Authors claim that flowering occurs one month after rainfall, however the graph appears to show that flowering peaks are placed two months after a rainfall peak. And the flowering peak in April 2011 is not preceded by strong precipitation as the one on June 2010. How much rain is it needed to trigger flowering? Can you disregard periodicity in flowering phenology? A less subjective form of analysis may shed some light on this.
Linear models (Lines 143, 159) should describe dependent and independent variables, as well as how they accounted for individual variation. Chi-square analysis are unable to accommodate nested designs, so authors need to clarify how they dealt with this issue. The degrees of freedom on F-tests (line 170) do not seem to agree with sample sizes since authors studied over 200 plants. Please clarify.

It is unclear how they accounted for variability in estimates of color variation. A multivariate or shape analysis should perhaps be considered to jointly analyse morphological variables. With that many variables, some are expected to vary between flowering episodes by chance alone, but these changes may not represent selective changes imposed by pollinators. Therefore, in the absence of pollinator information, significant differences in flower shape should be assessed.

Validity of the findings

The key findings of this study are: flowering episodes occur in the rainy season and may occur one (or two) months after a surge in precipitation; variability in flower morphology is attributed to individual level rather than flowering time. These results are unable to confirm if variation in floral shape between flowering episodes is linked to pollinator assemblages since they did not collect or identify pollinators. Therefore, we cannot determine the effect of pollinators or phenology on floral morphology and thus the authors are unable to reach conclusions about their original research question.

I agree that their results are useful for future studies and taxonomical descriptions.

The discussion could place their results in an evolutionary or ecological perspective. How are their results compared to similar species within the genus? Even if the manuscript is reframed as a descriptive study, these comparisons may make it interesting for a more general public. The discussion does not address the implications of individual variation nor heterostyly on reproductive success.
Authors should discuss if other factors (e.g., temperature) may be triggering flowering episodes.

Comments for the author

Several minor comments, questions or suggestions. Line numbers are used for reference.

21. In the abstract highly flowering individuals is confusing. Perhaps just say that flowering and fruiting was measured for 256 plants.
50. How does survival relate to phenology? It is unclear what you mean by this sentence. I think your argument is that abiotic factors may shape phenology.
70-75. This sentence is unclear. What do you mean by accurate configuration?
75. Heterostyly is not a diversification but rather an adaptation.
85-99. This is an interesting comparison of other Cordia species. This should perhaps be used to justify a descriptive analysis of C. boissieri. This should be also revisited inthe discussion.
107. It is unclear why seedlings are more likely to germinate when there are several reproductive events in contrast to a single large one. If one of those events occurs in an unsuitable time of the year, seeds will not germinate or recruit.
128. Later you mention that flowering only occurs during rainy season. How often is flowering in the dry season?
150. What defines a flowering event? Is it a certain proportion of individuals in the “high flowering” category?
152. Total length (TL) of what?
154. How many people performed nectar guide color estimation? How was this standarized? Direct color estimation is subject to variation by a lot of factors such as lightning conditions, gender and age of observer, among others. Did you account for any of these?
151, 156. What do you mean by style type?
158-163. Stats need to be explained in further detail.
170-174. A brief line describing what is considered a flowering event should be included. Synchronicity is usually estimated by looking at flowering overlap among individuals. How did you estimate synchronicity? Ambiguous use of language.
176. This paragraph is unclear, please consider rephrasing it.
183-184. There is specific terminology for heterostyly: pin-thrum or longistylus-brevistylous. Which one did you observe?
227. Late spring and early summer could be very close together. These are not seasons.
247. Discuss heterostyly in regards to the rest of the genus. Is this species closely related to others with similar flower morphology?
266. Still not clear how you estimated synchrony.

---

## Round 0.3 · accepted · Accept

· Academic Editor

Accept

Please review for additional minor typos/edits (e.g. "foe" instead of "of" on line 300).